# Resonant Valance Bond and Bethe Ansatz on Quasi-1D Lattices

**Zhao Zhang**[1*]

**1** Department of Physics, University of Oslo, P.O. Box 1048 Blindern, N-0316 Oslo, Norway

⋆ zhao.zhang@fys.uio.no

## Abstract

**The Hubbard model at $U \rightarrow \infty$ has recently been shown to have resonant valence bond (RVB) ground states on the corner-sharing sawtooth and pyrochlore lattices in the dilute doping limit of a single vacancy. In an effort to further generalize those results, I study how the ground state is modified when not all corners are shared between two tetrahedra as in the quasi-1D lattices of a pyrochlore stripe, and how to approach the problem in the case of finite doping. Using a non-Abelian version of the flux inequality, the tetrahedron chain is shown to have degenerate RVB-like ground states. The Bethe ansatz (BA) is adapted to solve the sawtooth chain with spinless or spin-polarized fermions and multiple holons, which is the first example of applying BA to a quasi-1D lattice.**

## 1 Introduction

The resonant valence bond (RVB) state is a competing ground state to Néel antiferromagnetic (AFM) order proposed by Anderson [1] in analogy to Pauling's attempted theory of metals

abandoning the Fermi gas picture. It was motivated by Bethe's solution of the spin-$\frac{1}{2}$ AFM Heisenberg chain [2], which shows a liquid-like ground state without sublattice magnetization in the absence of anisotropy. The hypothesis of quantum spin liquid regained interest following the discovery of high-temperature superconductivity, as doping allows the system to metalize and the existing real-space 'Cooper pairs' to Bose condense [3]. Short-range RVB states have been modeled by close-packed quantum dimers on square [4] and triangular lattices [5] under ring-exchange interactions, showing drastically different excitation spectra.[1] By construction, the ground states of quantum dimer models become frustration-free RVB states at the Rokhsar-Kivelson (R-K) point. However, the orthogonal dimer basis in the Hilbert space of dimer configurations are quite different from the overcomplete basis of spin singlets pairs formed by spin-$\frac{1}{2}$'s. While R-K moves can also be combined with Klein Hamiltonians [8] that projects onto non-dimerized SU(2) states to realize frustration-free spin Hamiltonians [9], the interaction has to involve a neighborhood of 8 spins on a square lattice.

Perhaps a more physically realistic route towards exact RVB state is via the exchange interaction of itinerant electrons instead of localized spin interactions. Itinerant ferromagnetism can appear with the infinite-$U$ Hubbard model at half-filling upon the doping of a single hole (or electron). If the expectation value of the hopping Hamiltonian is minimized by an antisymmetric spatial wavefunction between an electron and the holon, their spin configurations must be symmetric to satisfy fermionic statistics. Since the electron spins in the system are all antisymmetrized with the holon, their spins must be aligned with one other. This is the essence of Nagaoka ferromagnetism [10, 11], which applies to generic bipartite lattices. On a non-bipartite lattice, due to kinetic frustration, the spatial wavefunction of the electrons cannot be antisymmetrized. So a symmetric spatial wavefunction that minimizes the kinetic energy with negative hopping strength results instead in the AFM ordering of the spins [12,13]. Exact RVB ground states following this paradigm have been realized on the Husimi cacti [14, 15], checkerboard and pyrochlore lattices [16].

Although there is strong numerical evidence from exact diagonalization that the ground state of the two-hole doped pyrochlore lattice is still an equal superposition of spin-singlet dimer configurations with one dimer per tetrahedron [16], the result has not been analytically established. Moreover, little is known about how the ground state changes even when there are three vacancies in the lattice. The goal of the current paper is to understand how the RVB ground states change when things are slightly more complicated. First, the single-hole doped system is studied on the tetrahedron chain with spinful fermions. This can be viewed as a generalization to both the pyrochlore lattice and the sawtooth chain, where neither are all corners of a tetrahedron shared with a neighbor nor do the spin-doublet and quadruplet sectors of a hole-doped tetrahedron share the same Hamiltonian that facilitate the proof of the RVB ground state using the flux inequality [17]. Second, a momentum-space approach complementary to the previous real-space analyses is proposed to handle the many-body problem of holons in the simpler case of spinless fermions, which may provide insights into its spinful counterpart compatible with the RVB state when extended to nested Bethe ansatz.[2]

In one dimensional systems, Bethe's approach can be considered a generalized Fourier transform that maps quasiparticle excitations into the momentum space, where they scatter among each other exchanging momenta [2]. If the scattering matrices satisfy the consistency relation called the Yang-Baxter equation, the scattering among three or more quasi-

---

[1]It should be noted that the plaquette flipping move alone on the triangular lattice leads to Hilbert space fragmentation, resulting in exponentially large ground state degeneracy [6]. For bipartite lattices like the square lattice, due to the emergent Coulomb gauge, there is a well-defined height function on the dual lattice that leaves ground state degeneracy to be accounted for only by topological sectors [7]. This could be the reason behind the difference between a gapped and a gapless spectra.

[2]One should keep in mind that naively imposing periodic boundary condition on the sawtooth lattice would necessarily make the spin-singlet dimer bonds non-local.

particles is factorizable to sequences of two-body scattering. Furthermore, the scattering is non-diffractive, meaning eigenstates of the Hamiltonian are superposition of only different permutations of the momenta, as opposed to different values of the set of momenta. These two features of integrability is unique to 1D quantum systems, so the Bethe ansatz has to date not been applied to even quasi-1D lattices to the knowledge of the author. For that reason, this paper explores the finite doping effect first on simpler corner-sharing lattices of quasi-1D systems.

The plan for the rest of the article is the following, Sec. 2 generalizes the result on the RVB ground state in the sawtooth lattice to the pyrochlore stripe, after briefly reviewing the counter-Nagaoka effect. The second result based on Bethe ansatz is presented in Sec. 3, first for two-hole doped sawtooth lattice, which is then generalized to three holons and more. A conclusion including an outlook on the full nested Bethe ansatz treatment of the spinful fermion version of the finite doping problem of the model is given in Sec. 4.

## 2 RVB on Single-hole Doped Quasi-1D Lattices

### 2.1 Review of the RVB Ground State on the Sawtooth Lattice

The quasi-1D sawtooth lattice can be viewed as a single stripe of the 2D Kagome lattice, consisting of corner-sharing triangles. A unit cell has of two sites, a lower vertex $\alpha$ of degree 4 (except at the boundary of the chain), and an upper vertex $\beta$ of degree 2, and the Bravais lattice is labeled by an index $j \in 1, 2, \cdots, L$.[3] On each lattice site that is a vertex of the triangles, the local Hilbert space is spanned by three states, occupied by a fermion with spin up or down, or unoccupied. The interaction is that of the Fermi Hubbard model with infinite onsite potential

$$H = -t \sum_{j=1}^{L} \left( c_{j,\alpha}^\dagger c_{j,\beta} + c_{j,\alpha}^\dagger c_{j+1,\alpha} + c_{j,\beta}^\dagger c_{j+1,\alpha} + \text{h.c.} \right), \tag{1}$$

where the creation operators are dressed by projectors such that double occupancy is strictly forbidden.

Due to fermionic statistics and the existence of odd-length loops in the lattice, the effective hopping strength depends on the number of vacancies in the loop. The presence of a single holon introduces a $\pi$-flux that can be chosen to reverse the sign of the hopping strength across the horizontal bond. An effective hopping strength is specified by labeling the lattice sites and choosing the Hilbert space basis

$$|i, \boldsymbol{\sigma}\rangle = c_{1,\sigma_1}^\dagger c_{2,\sigma_1}^\dagger c_{3,\sigma_3}^\dagger \cdots c_{i-1,\sigma_{i-1}}^\dagger c_{i+1,\sigma_{i+1}}^\dagger \cdots c_{2L+1,\sigma_{2L+1}}^\dagger |\text{vac}\rangle. \tag{2}$$

Here, we label the lattice sites in ascending order from left to right, leading to the hopping constant for different holon configurations summarized in Fig. 1.

The single-hole doped many-body system can be approached by first understanding the few-body system of a single triangle, the ground state of which is the equal superposition of the holon located at three different corners with the fermions at the other two corners forming a spin singlet.[4] An effective hopping Hamiltonian for the holon can be written down, which is block diagonal, separating the subspaces of the two spins forming a single and a triplet. The Hamiltonians in the two sectors turn out to be identical up to a minus sign. The next

---

[3]Open boundary condition is chosen for this section, so there are $2L + 1$ vertices in the lattice. In Sec. 3, we will work instead with periodic boundary conditions.

[4]The dimers have a direction as opposite directions are off by a minus sign in $|\uparrow_j\downarrow_k\rangle - |\downarrow_j\uparrow_k\rangle$. The convention in this subsection is to always have the dimers pointing in the clockwise direction.

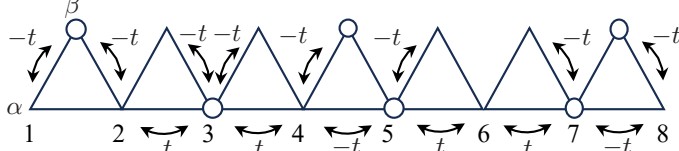

Figure 1: Effective hopping strength in the presence of vacancies.

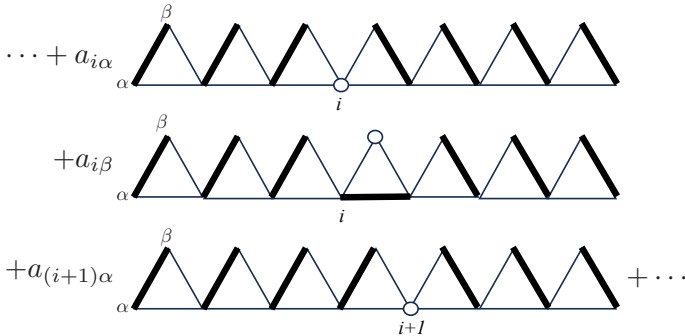

Figure 2: The ground state in the single holon sector as a superposition of domain wall configurations.

step is to use the flux inequality [17] to show that the lowest energy eigenstate resides in the sector where the dimers in all the triangles form spin singlets. Finally, the Perron-Frobenius theorem can be invoked to conclude that the ground state is a unique superposition of all dimer configurations [14]

$$
\begin{aligned}
|GS\rangle = \sum_{i=2}^{L-1} \Big( a_{i\alpha} \prod_{j=1}^{i-1} d^\dagger_{j\alpha,j\beta} d^\dagger_{i\beta,(i+1)\alpha} \prod_{k=i+1}^{L} d^\dagger_{k\beta,(k+1)\alpha} + a_{i\beta} \prod_{j=1}^{i-1} d^\dagger_{j\alpha,j\beta} d^\dagger_{(i+1)\alpha,i\alpha} \prod_{k=i+1}^{L} d^\dagger_{k\beta,(k+1)\alpha} \Big) |vac\rangle \\
+ \Big( a_{1\alpha} d^\dagger_{1\beta,2\alpha} + a_{i\beta} d^\dagger_{2\alpha,1\alpha} \Big) \prod_{k=2}^{L} d^\dagger_{k\beta,(k+1)\alpha} |vac\rangle + a_{(L+1)\alpha} \prod_{j=1}^{L} d^\dagger_{j\alpha,j\beta} |vac\rangle,
\end{aligned}
$$

$$(3)$$

where $d^\dagger_{u,v} = c^\dagger_{u,\uparrow} c^\dagger_{v,\downarrow} - c^\dagger_{u,\downarrow} c^\dagger_{v,\uparrow}$ represents a dimer singlet covering neighboring vertices $u$ and $v$, and $a_{iv}$ denotes the probability amplitude of the dimer configuration with the holon located at the vertex $v$ of the $i$th unit cell. The holon marks a domain wall between two segments with differently oriented dimers, as shown in Fig. 2.

## 2.2 RVB in Singly Doped Pyrochlore Stripe

Like the sawtooth lattice, the pyrochlore strip or tetrahedron chain has two types of vertices, one of degree 6, the other of degree 3. It has been studied previously under the Heisenberg interaction in Ref. [18], where under strong rung bonds, the ground state is shown to be a product of dimer states on the rung bonds and Bethe ansatz ground state of the Heisenberg chain. Although the model is different from the one of interest here, that result is also a consequence of the two different types of vertices. In this subsection, we examine how the results of Ref. [16] on the RVB ground state of single-hole doped 3D pyrochlore lattice get modified, when not all corners are shared between two tetrahedra.

An effective Hamiltonian of the infinite-$U$ Hubbard model on a tetrahedron with a single vacancy is determined by specifying a ordering of the vertices in the Fock basis, as shown in Fig. 2 (b). This Hamiltonian has been diagonalized in Ref. [16]. The 3-fold degenerate

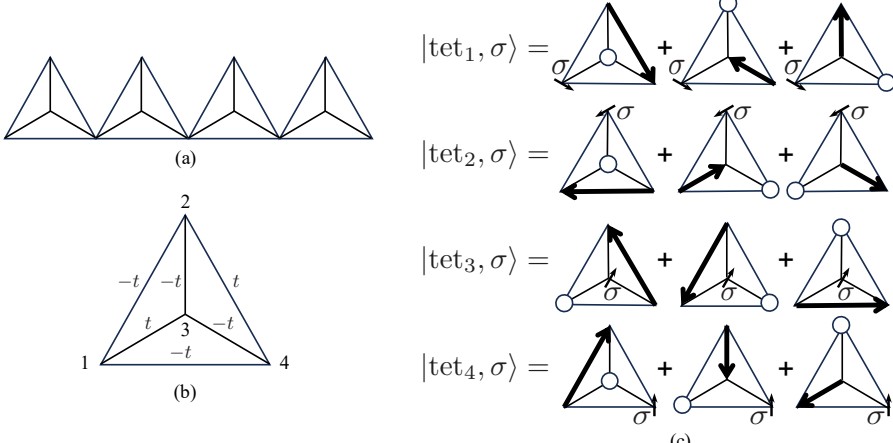

Figure 3: (a) A pyrochlore stripe consisting of 4 tetrahedra. (b) Choice of fermionic basis by assigning an order to the 4 vertices of a tetrahedron and the effective hopping constants. (c) The four degenerate ground states of a single tetrahedron. The large arrows represent a spin-singlet dimer and the small arrows represent a spin-doublet. Unlike in the sawtooth lattice which allows a fixed convention of the dimer direction, the direction of the dimers here are specified by arrows.

ground states (for $t > 0$) are presented in Fig. 2 (c), along with their linear combination $|\text{tet}_4\rangle = -\sum_{i=1}^{3} |\text{tet}_i\rangle$. Notice that they have not been orthonormalized. The four ground states are labeled by the location of the spin-doublet. In the face opposite to the doublet, the holon and the dimer form the same ground state of as the triangle in the previous subsection. But $|\text{tet}_{1,4}\rangle$ is different from $|\text{tet}_{2,3}\rangle$ in that the triangle has two corners on the lattice sites of the 1D chain, instead of one. Heuristically speaking, this results in degenerate ground states in the tetrahedron chain.

Because the hopping Hamiltonian commutes with both the total spin operator $S$ of the three spins in a tetrahedron with the holon and its $S_z$ component, their eigenvalues are both good quantum numbers. The tensor product of three spin-$\frac{1}{2}$ representations of SU(2) is decomposed into a spin-$\frac{3}{2}$ quadruplet and two spin-$\frac{1}{2}$ doublets

$$\frac{1}{2} \otimes \frac{1}{2} \otimes \frac{1}{2} = \left(0 \oplus 1\right) \otimes \frac{1}{2} = \frac{3}{2} \oplus \frac{1}{2} \oplus \frac{1}{2}.$$

Without loss of generality, we can look at the sector ↑↑↓ with two spin-ups and one spin-down. Together with the location of the holon and the pairing of the dimer, this subspace is 8-dimensional spanned by the orthonomal basis shown in Fig. 4 (a). The matrix representation of the effective Hamiltonian in this basis is given as

$$H^{\text{db}} = \begin{pmatrix} 0_{2\times2} & H_1 & H_2 & H_3 \\ H_1^T & 0_{2\times2} & H_1 & H_2 \\ H_2^T & H_1^T & 0_{2\times2} & H_1 \\ H_3^T & H_2^T & H_1^T & 0_{2\times2} \end{pmatrix}, \tag{4}$$

with

$$H_1 = t\begin{pmatrix} \frac{1}{2} & \frac{\sqrt{3}}{2} \\ -\frac{\sqrt{3}}{2} & \frac{1}{2} \end{pmatrix}, \quad H_2 = t\begin{pmatrix} \frac{1}{2} & -\frac{\sqrt{3}}{2} \\ -\frac{\sqrt{3}}{2} & -\frac{1}{2} \end{pmatrix}, \quad H_3 = t\begin{pmatrix} \frac{1}{2} & -\frac{\sqrt{3}}{2} \\ \frac{\sqrt{3}}{2} & \frac{1}{2} \end{pmatrix}. \tag{5}$$

$H^{\text{db}}$ has eigenvalues $\pm 2t$ (three-fold degenerate) and $0$ (doubly degenerate). The Hamiltonian

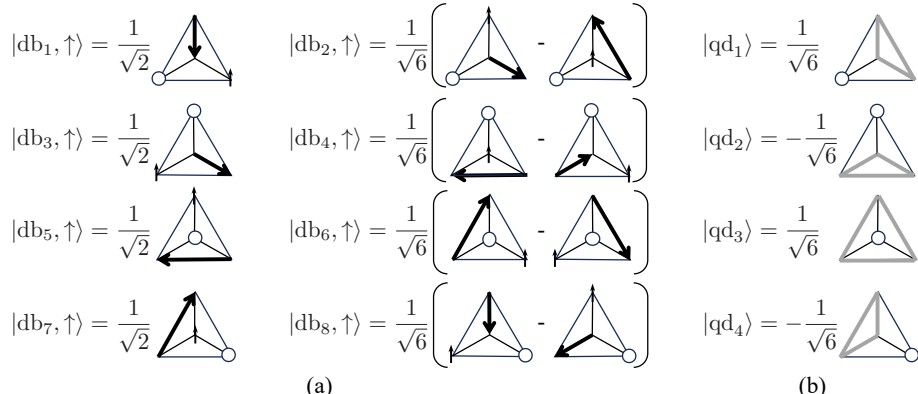

Figure 4: An orthonomal basis for the subspace with two spin-ups and one spin-down in (a) the 8-dimensional spin-doublet sector, and (b) the 4-dimensional spin-quadruplet sector. The grey triangle represents that the spins at the three corners are symmetrized.

in the basis of the quadruplet subspace shown in Fig. 4 (b) is instead

$$
H^{\mathrm{qd}} = t \begin{pmatrix} 0 & 1 & 1 & 1 \\ 1 & 0 & 1 & 1 \\ 1 & 1 & 0 & 1 \\ 1 & 1 & 1 & 0 \end{pmatrix},
\tag{6}
$$

with eigenvalues $3t$ and $-t$ (three-fold degenerate).

So we have two sets of basis to choose from for three spins in each tetrahedron of the chain, with different hopping Hamiltonians in different bases.[5] But the lowest energy eigenstate must be in the subspace where the three spins in each tetrahedron form spin-doublets simultaneously. The proof of this claim is a non-Abelian version of the flux inequality used in Ref. [14], which goes as follows.

Suppose that $|\psi\rangle$ is the normalized lowest energy eigenstate in the subspace where the three spins in each tetrahedron form a quadruplet. Because the effective Hamiltonian has only non-negative matrix elements, its eigenvectors can all be chosen to have real components $\psi_j$, for $1 \leq j \leq 3L+1$. $\psi_j$ corresponds to the configuration with the holon at site $j$, which are ordered within each tetrahedron according to Fig. 3 (b). Furthermore, due to the symmetry between sites $3k-1$ and $3k$, the eigenstates can be chosen such that $\psi_{3k-1} = \pm\psi_{3k}$, for $1 \leq k \leq L$. It is easy to see that the symmetrized state can always be made lower energy than the antisymmetrized one.[6] The energy of the symmetrized state is the sum of the expectation value of the Hamiltonian (6) within each tetrahedron

$$
E = 2t \sum_{k=1}^{L} \left( 2\psi_{3k-2}\psi_{3k-1} + 2\psi_{3k-1}\psi_{3k+1} + \psi_{3k-2}\psi_{3k+1} + \psi_{3k-1}^2 \right).
\tag{7}
$$

Now we can use that fact to construct a state $|\psi'\rangle$ in the sector where the three spins in each tetrahedron form instead a spin-doublet that has lower energy. $|\psi'\rangle$ has twice as many components because there are two orthogonal states of the spin configuration for the

---

[5]Except at most two neighboring tetrahedra that host the holon, all the others have four spins instead. Of course, what is meant here is that the spins are grouped in groups of threes starting from the location of the holon.

[6]Indeed, instead of the summand in (7), the antisymmetrized state would have $-\psi_{3k+2}^2 + \psi_{3k+1}\psi_{3k+4}$. Their difference $2\psi_{3k-1}(\psi_{3k-1} + \psi_{3k-2} + \psi_{3k+1})$ is negative as long as $|\psi_{3k-2} + \psi_{3k+1}| > |\psi_{3k-1}|$ and the two have the opposite signs, which is possible while keeping the state normalized.

same location of the holon, as shown in Fig. 4. They can be chosen as $\psi'_{2j-1} = |\psi_j| \cos \theta_j$, and $\psi'_{2j} = |\psi_j| \sin \theta_j$, such that it is still a normalized state. The energy of this new state $E' = \langle \psi' | \sum_{k=1}^{L} H_k^{db} | \psi' \rangle$ can be expressed as

$$
E' = 2t \sum_{k=1}^{L} \Big( 2\cos(\theta_{3k-2} + \frac{\pi}{3}) \cos \theta_{3k-1} |\psi_{3k-2} \psi_{3k-1}| + 2\cos(\theta_{3k-1} + \frac{\pi}{3}) \cos \theta_{3k+1} |\psi_{3k-1} \psi_{3k+1}|
$$
$$
+ \cos(\theta_{3k+1} - \theta_{3k-2} + \frac{\pi}{3}) |\psi_{3k-2} \psi_{3k+1}| + \cos(\theta_{3k-1} - \theta_{3k} + \frac{\pi}{3}) \psi_{3k-1}^2 \Big).
$$
(8)

To state the obvious, the first three terms in the summand of (7) cannot simultaneously be negative. In fact, for $E$ to be the lowest energy state of its sector, $\psi_{3k-2}$ and $\psi_{3k+1}$ must have opposite signs.[7] In order to make $E' \leq E$, we just need to let the cosine factors in front of the third and one of the first two terms in (8) be $-1$ simultaneously. For instance, if $\psi_{3k-2} \psi_{3k-1} < 0$, we need

$$
\begin{cases}
\cos(\theta_{3k-2} + \dfrac{\pi}{3}) \cos \theta_{3k-1} = -1, \\
\cos(\theta_{3k+1} - \theta_{3k-2} + \dfrac{\pi}{3}) = -1.
\end{cases}
$$
(9)

One solution $\theta_{3k-2} = \frac{2\pi}{3}, \theta_{3k-1} = 0, \theta_{3k+1} = \frac{4\pi}{3}$ makes the cosine factors of the second term $-\frac{1}{4}$, which suffices our purpose.

It only takes slight modifications for the proof above to work also when the state we started with $|\psi\rangle$ has only part of the tetrahedra in the spin-quadruplet sector.[8] So the conclusion is that in the ground state, all the tetrahedra in the chain must form spin-doublets, meaning that the ground states are superpositions of configurations with $L$ dimers, one holon, and $L$ spin-doublet monomers. However, unlike the sawtooth case, because the matrix elements of (4) are not all negative, we cannot deduce that the ground state is unique. This ground state degeneracy is essentially attributed to the degenerate ground states of the single tetrahedron. On top of that, the total $z$-component of the spin is also a good quantum number, which does not affect the ground state energy as long as the total spin within each tetrahedron is $\frac{1}{2}$.

At finite $U$ of the Hubbard model, the AFM Heisenberg interaction of the effective $t-J$ model is expected to lift at least some of the degeneracy. However, unlike the sawtooth case, its lowest energy on a single tetrahedron is achieved when the four spins form two singlet dimers. Thus for a chain of tetrahedra, the ground state of the AFM Heisenberg interaction is frustrated, making it also compete with the hopping Hamiltonian.

## 3   Quasi-1D Bethe Ansatz for Spinless Fermions

In the previous section, although we were able to conclude that the ground states of the sawtooth and tetrahedron chains are RVB states (with additional monomers for the latter), the weighting among dimer configurations are not computed analytically. This is mostly due to the open boundary condition which breaks the full translational invariance. Changing to periodic boundary condition would either introduce additional holons or spin-doublet monomers

---

[7]Without loss of generality, suppose that $\psi_{3k-1} > 0$, $\psi_{3k-2} < 0$, and $\psi_{3k+1} < 0$, then by flipping the signs of $\psi_{3k-2}$, the energy expectation value would be $2t \sum_{k=1}^{L} \psi_{3k-2}(2\psi_{3k-1} + \psi_{3k+1})$ lower, which is positive as long as $\psi_{3k+1} < -2\psi_{3k-1}$.

[8]Had we started with a state where the $(k+1)$th tetrahedron was a spin-doublet while the $k$th was a quadruplet, the energy for the $k$th tetrahedron would just consist of two copies of the expression in (7), for each of which the same arguments can be used to show that the energy would be lower if the $k$th tetrahedron was also a spin-doublet.

Figure 5: Basis vectors in the ansatz eigenstates in the two-holon sector.

in an otherwise dimerized spin-singlet paring. Another consequence is that the dimers would inevitably become delocalized or long-range as a holon can be hopping from either side of a dimer.

For coordinate Bethe ansatz to be applicable, the local Hilbert space of a lattice site has to be two dimensional, so in this section, we work with spinless fermions on the sawtooth lattice. Alternatively, they can be considered spin polarized fermions. It also happens to be the subspace to which the ground state of a $t < 0$ model belongs. It is possible that the spinful fermion problem can be solved similarly in the framework of nested Bethe ansatz, but that will the topic of future works.

## 3.1 The Two-body Problem

Using translational invariance, we can diagonalize the Hamiltonian within each subspace with a given eigenvalue of the translation operator. The ansatz due to Bethe is that each holon travels as a plane wave with momenta $\mu_{1,2}$, and an additional phase factor is imposed when they scatter with each other. For our quasi-1D lattice, additional phase factors can arise when the holon hops between the two sublattices. Hence, an eigenstate can be generically expressed as

$$|\mu_1, \mu_2\rangle = \sum_{j,k=1}^{L} \sum_{\sigma \in \mathcal{S}_2} \sum_{v \in \{\alpha, \beta\}^{\otimes 2}} A_\sigma^v \mu_{\sigma 1}^j \mu_{\sigma 2}^k |(j, v_1), (k, v_2)\rangle, \tag{10}$$

where the second sum is over permutations between indices 1 and 2, and the basis vectors are graphically represented in Fig. 5.

Acting (1) on (10), we obtain 4 eigenvalue equations, one for the component of each basis vector. Their explicit forms can vary a little depending on whether $j$ and $k$ are adjacent or not, as some scattering terms are nonexistent when they are. For (10) to be an eigenstate, both types of equations need to be simultaneously satisfied. So below we just list the eigenvalue equations when $j$ and $k$ are sufficiently apart, and the difference between two types of equations as independent constraints to be satisfied.

For the component in $|(j, \alpha), (k, \alpha)\rangle$, we have

$$E_2 \sum_{\sigma \in \mathcal{S}_2} A_\sigma^{(\alpha,\alpha)} \mu_{\sigma 1}^j \mu_{\sigma 2}^k = t \sum_{\sigma \in \mathcal{S}_2} \Big[ A_\sigma^{(\alpha,\alpha)} \sum_{\delta = \pm 1} \big( \mu_{\sigma 1}^{j+\delta} \mu_{\sigma 2}^k + \mu_{\sigma 1}^j \mu_{\sigma 2}^{k+\delta} \big)$$
$$- \sum_{\epsilon = 0,1} \big( A_\sigma^{(\alpha,\beta)} \mu_{\sigma 1}^j \mu_{\sigma 2}^{k-\epsilon} + A_\sigma^{(\beta,\alpha)} \mu_{\sigma 1}^{j-\epsilon} \mu_{\sigma 2}^k \big) \Big], \quad j+1 < k; \quad (11)$$

$$A_{12}^{(\alpha,\alpha)} \mu_1^{j+1} \mu_2^k + A_{21}^{(\alpha,\alpha)} \mu_2^{j+1} \mu_1^k + A_{12}^{(\alpha,\alpha)} \mu_1^j \mu_2^{k-1} + A_{21}^{(\alpha,\alpha)} \mu_2^j \mu_1^{k-1} = 0, \qquad j+1 = k. \quad (12)$$

Unless $\mu_1 \mu_2 = -1$, the latter is only satisfied if $A_{21}^{(\alpha,\alpha)} = -A_{12}^{(\alpha,\alpha)}$. This in turn simplifies (11) to

$$\Big[ t\big(\mu_1 + \mu_1^{-1} + \mu_2 + \mu_2^{-1}\big) - E_2 \Big] A_{12}^{(\alpha,\alpha)} \big(\mu_1^j \mu_2^k - \mu_2^j \mu_1^k\big) = t \Big[ \big(1 + \mu_2^{-1}\big) A_{12}^{(\alpha,\beta)} + \big(1 + \mu_1^{-1}\big) A_{12}^{(\beta,\alpha)} \Big] \mu_1^j \mu_2^k$$
$$+ t \Big[ \big(1 + \mu_1^{-1}\big) A_{21}^{(\alpha,\beta)} + \big(1 + \mu_2^{-1}\big) A_{21}^{(\beta,\alpha)} \Big] \mu_2^j \mu_1^k,$$

which only holds if $A_{21}^{(\beta,\alpha)} = -A_{12}^{(\alpha,\beta)}$, $A_{21}^{(\alpha,\beta)} = -A_{12}^{(\beta,\alpha)}$, and

$$\left[E_2 - t\left(\mu_1 + \mu_1^{-1} + \mu_2 + \mu_2^{-1}\right)\right]A_{12}^{(\alpha,\alpha)} + t\left(1 + \mu_2^{-1}\right)A_{12}^{(\alpha,\beta)} + t\left(1 + \mu_1^{-1}\right)A_{12}^{(\beta,\alpha)} = 0. \quad (13)$$

For the component in $|(j,\alpha),(k,\beta)\rangle$, we have

$$E_2 \sum_{\sigma \in \mathcal{S}_2} A_\sigma^{(\alpha,\beta)} \mu_{\sigma 1}^j \mu_{\sigma 2}^k = t \sum_{\sigma \in \mathcal{S}_2} \left[A_\sigma^{(\alpha,\beta)} \sum_{\delta = \pm 1} \mu_{\sigma 1}^{j+\delta} \mu_{\sigma 2}^k \right.$$
$$\left. - \sum_{\epsilon = 0,1} \left(A_\sigma^{(\alpha,\alpha)} \mu_{\sigma 1}^j \mu_{\sigma 2}^{k+\epsilon} + A_\sigma^{(\beta,\beta)} \mu_{\sigma 1}^{j-\epsilon} \mu_{\sigma 2}^k\right)\right], \quad j < k; \quad (14)$$

$$\sum_{\sigma \in \mathcal{S}_2} \left[A_\sigma^{(\alpha,\beta)} \mu_{\sigma 1}^{j+1} \mu_{\sigma 2}^k + A_\sigma^{(\beta,\alpha)} \mu_{\sigma 1}^j \mu_{\sigma 2}^{k+1} - \left(A_\sigma^{(\alpha,\alpha)} + A_\sigma^{(\beta,\beta)}\right)\mu_{\sigma 1}^j \mu_{\sigma 2}^k\right] = 0, \qquad j = k. \quad (15)$$

Given the relations we already obtained from (12), (15) implies $A_{21}^{(\beta,\beta)} = -A_{12}^{(\beta,\beta)}$.

For the component in $|(j,\beta),(k,\alpha)\rangle$, we have

$$E_2 \sum_{\sigma \in \mathcal{S}_2} A_\sigma^{(\beta,\alpha)} \mu_{\sigma 1}^j \mu_{\sigma 2}^k = t \sum_{\sigma \in \mathcal{S}_2} \left[A_\sigma^{(\beta,\alpha)} \sum_{\delta = \pm 1} \mu_{\sigma 1}^j \mu_{\sigma 2}^{k+\delta} \right.$$
$$\left. - \sum_{\epsilon = 0,1} \left(A_\sigma^{(\alpha,\alpha)} \mu_{\sigma 1}^{j+\epsilon} \mu_{\sigma 2}^k + A_\sigma^{(\beta,\beta)} \mu_{\sigma 1}^j \mu_{\sigma 2}^{k-\epsilon}\right)\right], \quad j+1 < k; \quad (16)$$

$$\sum_{\sigma \in \mathcal{S}_2} \left[\left(A_\sigma^{(\alpha,\beta)} + A_\sigma^{(\beta,\alpha)}\right)\mu_{\sigma 1}^j \mu_{\sigma 2}^{k-1} - A_\sigma^{(\alpha,\alpha)} \mu_{\sigma 1}^{j+1} \mu_{\sigma 2}^k - A_\sigma^{(\beta,\beta)} \mu_{\sigma 1}^j \mu_{\sigma 2}^{k-1}\right] = 0, \qquad j+1 = k. \quad (17)$$

Now (17) is already ensured by (12) and (15).

Finally, for the component in $|(j,\beta),(k,\beta)\rangle$, we have

$$E_2 \sum_{\sigma \in \mathcal{S}_2} A_\sigma^{(\beta,\beta)} \mu_{\sigma 1}^j \mu_{\sigma 2}^k = -t \sum_{\sigma \in \mathcal{S}_2} \sum_{\epsilon = 0,1} \left[A_\sigma^{(\alpha,\beta)} \mu_{\sigma 1}^{j+\epsilon} \mu_{\sigma 2}^k + A_\sigma^{(\beta,\alpha)} \mu_{\sigma 1}^j \mu_{\sigma 2}^{k+\epsilon}\right], \qquad j < k. \quad (18)$$

Using the relations between the amplitudes derived from the meeting equations, (14),(16) and (18) can be simplified as

$$t\left(1 + \mu_2\right)A_{12}^{(\alpha,\alpha)} + \left[E_2 - t\left(\mu_1 + \mu_1^{-1}\right)\right]A_{12}^{(\alpha,\beta)} + t\left(1 + \mu_1^{-1}\right)A_{12}^{(\beta,\beta)} = 0,$$
$$t\left(1 + \mu_1\right)A_{12}^{(\alpha,\alpha)} + \left[E_2 - t\left(\mu_2 + \mu_2^{-1}\right)\right]A_{12}^{(\beta,\alpha)} + t\left(1 + \mu_2^{-1}\right)A_{12}^{(\beta,\beta)} = 0,$$
$$t\left(1 + \mu_1\right)A_{12}^{(\alpha,\beta)} + t\left(1 + \mu_2\right)A_{12}^{(\beta,\alpha)} + E_2 A_{12}^{(\beta,\beta)} = 0.$$

Together with (13), there is only a non-trivial solution of amplitudes if the determinant of their coefficient matrix vanishes, which happens at

$$E_2 = t\left(\cos\theta_1 + \cos\theta_2 \pm \sqrt{(\cos\theta_1 + 1)^2 + 1} \pm \sqrt{(\cos\theta_2 + 1)^2 + 1}\right), \qquad (19)$$

where $\theta_{1,2} = -i \ln \mu_{1,2}$ are the momenta of the two quasiparticles. Their values can be determined from the periodic boundary condition $\mu_{1,2}^L = -1$. The energy difference from that of two plane waves decomposes into two parts that each depends on only one momentum. This means that the two holons scatter with one another like free fermions, but their energy is dressed by the self interaction within a unit cell.

## 3.2 The Three-body problem

When we write down the 3-holon trial wavefunction, we can take advantage of the scattering phases we already know from two-body scatterings, namely two holons in the same sublattice passing around each other induces a $\pi$ phase difference. So we can write

$$
\begin{aligned}
|\mu_1, \mu_2\rangle = \sum_{j,k,l=1}^{L} \Big[ &\sum_{\sigma \in \mathcal{S}_3} (-1)^\sigma \mu_{\sigma 1}^j \mu_{\sigma 2}^k \mu_{\sigma 3}^l \big(A|j\alpha, k\alpha, l\alpha\rangle + D|j\beta, k\beta, l\beta\rangle\big) \\
&+ \sum_{\rho \in \mathcal{S}_2} (-1)^\rho \Big( \mu_1^j \mu_{\rho 2}^k \mu_{\rho 3}^l \big(B_1|j\beta, k\alpha, l\alpha\rangle + C_1|j\alpha, k\beta, l\beta\rangle \\
&+ \mu_{\rho 1}^j \mu_2^k \mu_{\rho 3}^l \big(B_2|j\alpha, k\beta, l\alpha\rangle + C_2|j\beta, k\alpha, l\beta\rangle\big) \\
&+ \mu_{\rho 1}^j \mu_{\rho 2}^k \mu_3^l \big(B_3|j\alpha, k\alpha, l\beta\rangle + C_3|j\beta, k\beta, l\alpha\rangle\big)\Big) \Big],
\end{aligned}
\tag{20}
$$

where $(-1)^\sigma$ denotes the parity of the permutation $\sigma$. The remaining scattering phases are determined from the following equation

$$
\begin{pmatrix}
\frac{E_3}{t}-\epsilon_0 & 1+\mu_1^{-1} & 1+\mu_2^{-1} & 1+\mu_3^{-1} & 0 & 0 & 0 & 0 \\
1+\mu_1 & \frac{E_3}{t}-\epsilon_1 & 0 & 0 & 0 & 1+\mu_3^{-1} & 1+\mu_2^{-1} & 0 \\
1+\mu_2 & 0 & \frac{E_3}{t}-\epsilon_2 & 0 & 1+\mu_3^{-1} & 0 & 1+\mu_1^{-1} & 0 \\
1+\mu_3 & 0 & 0 & \frac{E_3}{t}-\epsilon_3 & 1+\mu_2^{-1} & 1+\mu_1^{-1} & 0 & 0 \\
0 & 0 & 1+\mu_3 & 1+\mu_2 & \frac{E_3}{t}-\epsilon_4 & 0 & 0 & 1+\mu_1^{-1} \\
0 & 1+\mu_3 & 0 & 1+\mu_1 & 0 & \frac{E_3}{t}-\epsilon_5 & 0 & 1+\mu_2^{-1} \\
0 & 1+\mu_2 & 1+\mu_1 & 0 & 0 & 0 & \frac{E_3}{t}-\epsilon_6 & 1+\mu_3^{-1} \\
0 & 0 & 0 & 0 & 1+\mu_1 & 1+\mu_2 & 1+\mu_3 & \frac{E_3}{t}
\end{pmatrix}
\begin{pmatrix} A \\ B_1 \\ B_2 \\ B_3 \\ C_1 \\ C_2 \\ C_3 \\ D \end{pmatrix}
=
\begin{pmatrix} 0 \\ 0 \\ 0 \\ 0 \\ 0 \\ 0 \\ 0 \\ 0 \end{pmatrix}
\tag{21}
$$

where $\epsilon_0 = \sum_{j=1}^3 (\mu_j + \mu_j^{-1})$, $\epsilon_k = \epsilon_0 - (\mu_k + \mu_k^{-1})$ for $k = 1, 2, 3$, and $\mu_{k-4} + \mu_{k-4}^{-1}$ for $k = 5, 6, 7$. Like before, the energy eigenvalues $E_3$ are still determined by the vanishing of the determinant of the coefficient matrix, giving

$$
E_3 = t \sum_{i=1}^{3} \big( \cos\theta_i \pm \sqrt{(\cos\theta_i + 1)^2 + 1} \big),
\tag{22}
$$

but the momenta themselves satisfy the new periodic boundary condition $\mu_{1,2}^L = 1$.

The difference from conventional Bethe Ansatz is that due to the quasi-1D nature of the lattice, the scattering of the Hamiltonian creates multiple outcome states. Thus instead of a scattering phase dependent purely on the momenta of the two quasiparticles involved, all the scattering phases are determined together by all of the momenta. So the complexity of the eigenvalue equation grows exponentially with the number of quasiparticles, which is still a major shortcut than diagonalizing the full Hamiltonian, which grows exponentially with the system size. On the other hand, the consistency relations imposed by the Yang-Baxter equation does not play a role here, so we do not need to worry about the integrability of the Hamiltonian.

## 4 Conclusion

In this article, two possible extensions to the recently discovered RVB ground states in triangle cacti and the pyrochlore lattice are explored, one concerning the modification when not all corners of the tetrahedra are shared, the other the multiple-holon scenario with periodic boundary conditions. The first result generalizes that of the same model on the sawtooth lattice to a ground state with resonating mixture of dimers and monomers, except at finite $U$ or

non-vanishing $J$. The second result is restricted to spinless fermions, but is also relevant to the low-energy sector of itinerant ferromagnetism.

The two results are obtained with complementing approaches, one from real-space analyses within a local tetrahedron, the other from the momentum space based on Bethe ansatz. It is therefore interesting to see if there a interpolating method that combines the advantage of both of them, before attempting to apply the nested Bethe ansatz to the finite doping problem, which completely hides the dimer structure in eigenstates. One related construction is the partially integrable states of $S_N$ symmetric chains, which is solved by applying the coordinate Bethe ansatz to a background of pre-antisymmetrized basis of $S_n$ singlets [19]. Another successful example is the application of Bethe ansatz on top of a matrix product state and the generalized frustration-free model [20].

An alternative to using the periodic boundary condition is to choose integrable reflecting boundary conditions that allows the probability amplitudes in the RVB ground state to be solved analytically. This is also compatible with single vacancy, and perhaps simpler than the treatment in Sec. 3. Since little is known about quantum integrability on quasi-1D lattices, it is worth investigating integrability and free fermion conditions on these lattices. In particular, whether there exist infinitely many local conserved charges and if there is any algebraic structure among them [21].

Last but not least, it is interesting to see if there are possible generalizations on lattices composed of higher simplex structures beyond triangles and tetrahedra. One can also consider more local degrees of freedom. Along these lines, the results of diagonalizing the SU($k$) version of the XX model on $k$-simplices might become relevant [22].

## Acknowledgements

I thank Cecilie Glittum and Olav Syljuåsen for introducing me to kinetic frustration and doping induced RVB state and for their valuable feedback on the manuscript.

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
