# Peer review of "Resonant Valance Bond Ground States on Corner-sharing Lattices"

_SciPost Physics_

## Round 2 · Referee Report · Anonymous (Referee 1) · 2026-1-7

Disclosure of Generative AI use

The referee discloses that the following generative AI tools have been used in the preparation of this report:

ChatGPT 5.2 was used (7. January) to check gramma and spelling of my comment, since I am no native speaker.

Report

The authors study the infinite-U Hubbard model on a one-dimensional pyrochlore stripe with a single-hole excitation. This setup can be viewed as a one-dimensional analogue of the two-dimensional pyrochlore lattice with a single hole, previously investigated in Ref. [10]. To analyze the model, the authors generalize and apply methods from Ref. [15], where the infinite-U Hubbard model on the sawtooth chain was studied. In particular, they employ an adapted version of the diamagnetic inequality to determine the symmetry sector in which the ground states reside, and they construct the exact ground states in the thermodynamic limit within that sector. The resulting ground states describes a hole moving in a resonating-valence-bond background, in close analogy to the findings of Ref. [10].
Overall, the paper is highly relevant, as it may open a route to understanding the model studied in Ref. [10] from a different perspective and could guide investigations of other one-dimensional models with similar features. The manuscript is well structured and clearly written; nevertheless, I have a few minor comments and suggested changes, as detailed below.

Requested changes

1- On page 3, in footnote 2, the last sentence appears to be incomplete. 2- I do not see how the normalization factors $\dfrac{1}{\sqrt{6}}$ in Fig. 5 are obtained. 3-I would not object to renaming the “flux inequality” as the “diamagnetic inequality,” since this is the terminology most commonly used in the literature, including in the cited references. 4-The manuscript states that inequality (20) is “obviously” true because inequality (21) holds. However, it is not clear to me (i) why inequality (21) holds and (ii) how (20), in its squared form, follows from it. A few sentences clarifying these steps would be useful.

Recommendation

Publish (meets expectations and criteria for this Journal)

---

## Editorial Decision

in_refereeing